# *PFAvatar*: Avatar Reconstruction from Multiple In-the-wild Images

## Abstract

In this work, we present *PFAvatar*, a new approach to avatar reconstruction and editing from multiple in-the-wild images with varying poses, unknown camera conditions, cropped views, and occlusions. Traditional methods often rely on full-body images captured with controlled avatar pose, camera settings, lighting, and background, while struggling to reconstruct under in-the-wild settings. To address this issue, we fuse the varying pose priors of avatars in in-the-wild images, thereby enabling precise control over avatar generation. Specifically, we first inject avatar features (pose, appearance) from input images using a Vision-Language Model (VLM) and ControlNet. Subsequently, we employ a pose-conditioned 3D-Consistent Score Distillation Sampling (3D-SDS), which enables reconstructing a high-quality 3D avatar. Additionally, we propose a Condition Prior Preservation Loss (CPPL) to mitigate the issues of language and control drift caused by fine-tuning VLM and ControlNet with few-shot data. Through comprehensive experiments and evaluation, we demonstrate the effectiveness of our method for reconstructing avatars from in-the-wild images, supporting further applications like avatar editing.

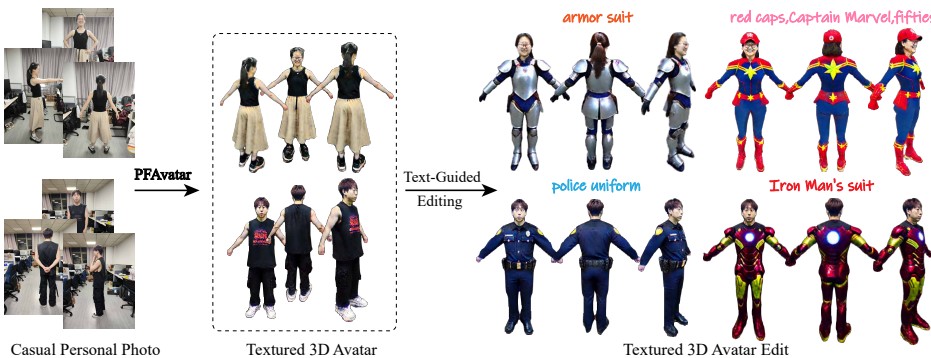

Figure 1: With just a few images of a personal casual photo (left), **PFAvatar** reconstructs a faithful, personalized, and textured 3D avatar from a personal photo collection (middle). These images can vary in body poses, camera angles, framing, lighting, and backgrounds. **PFAvatar** also supports downstream tasks, such as customizing avatars and performing virtual try-on via Text-Guided Editing, while preserving the subject's identity (right).

## 1 Introduction

The creation of 3D human avatars from texts or images has long been a challenging problem in computer vision and graphics, which is crucial for various applications such as digital humans, the film industry, and virtual reality. Although text-guided (Liu et al., 2023; Sun et al., 2023; Cao et al., 2024; Zeng et al., 2024; Poole et al., 2022) digital human generation has made substantial progress in creating avatars of well-known characters (e.g., Spider-Man), generating avatars with casual capture setups remains a difficult challenge.

Traditional approaches (Alexander et al., 2010; Guo et al., 2017; Xiong et al., 2024; Shen et al., 2023; Işık et al., 2023) typically depend on full-body images captured under controlled environments, with

strict requirements for avatar pose, camera settings, lighting, and background. Additionally, they often require multi-view images or depth maps, which are impractical for consumer-level applications. Alternatively, other methods leverage a neural network to predict plausible avatar models from a single image or video input (Habermann et al., 2020; Yang et al., 2023; Xiu et al., 2022; Zheng et al., 2020; Zhang et al., 2024c) but perform poorly under in-the-wild situations, such as unusual body poses, motion blur, and occlusions, because they rely on accurate human and camera pose estimation from full-body shots. In daily life, we usually only have access to few-shot in-the-wild images obtained by phone cameras, with varying poses, unknown camera settings, cropped views, and random occlusions of individuals. Thus, a method that can accurately reconstruct 3D human avatars from few-shot in-the-wild images will significantly cut costs and simplify the process of independent creation.

Reconstructing avatars from in-the-wild images is difficult for two reasons. The first is the absence of exact pixel-wise correspondence between the input images and the reconstructed avatars. Existing avatar reconstruction methods require projecting image features onto 3D avatars (Alldieck et al., 2022; Xiu et al., 2022; Cao et al., 2023b; Corona et al., 2023; Saito et al., 2019; 2020; Yang et al., 2023) or employing fixed learnable embeddings to generate 3D features (Zhang et al., 2023c). The absence of 3D correspondences makes directly fusing 3D information from the images a highly challenging task. The second is caused by the sparse and irregular viewpoint. High-fidelity 3D representations require a large number of input images (Zou et al., 2023; Zhang et al., 2021; Goel et al., 2022; Zhou & Tulsiani, 2023; Long et al., 2022; Cerkezi & Favaro, 2024) and have difficulty in handling sparse viewpoints. Meanwhile, current pose estimation from sparsely sampled views (Wang et al., 2023c; Zhang et al., 2024b; 2022; Lin et al., 2023a; Wang et al., 2023b) typically requires the avatar pose to remain fixed. In short, it is still a challenging task to reconstruct a high-quality avatar from a set of in-the-wild images.

In this paper, in response to these challenges, we introduce a novel method as shown by Figure 2, dubbed **PFAvatar**, for avatar reconstruction and editing using multiple in-the-wild images. Our insight is to treat vision-language models (Rombach et al., 2021; Ruiz et al., 2023) and T2I generation models as personalized priors, which allows us to avoid the need for explicit per-pixel correspondences to a canonical human space while also bypassing camera pose estimation. These Text-to-Image (T2I) generation methods (Gao et al., 2023; Zhang et al., 2024c; Wu et al., 2023; Yang et al., 2024; Zhang et al., 2023a) treat reconstruction from partial observations as a process of "inpainting" unobserved regions using foundational-model priors, enforcing cross-view consistency. Then, the Score Distillation Sampling (SDS) (Poole et al., 2022) is further proposed to boost the performance by distilling the 2D knowledge from a pre-trained diffusion model (Ho et al., 2020; Rombach et al., 2022; Hong et al., 2022a) to 3D content generation via differentiable rendering. Additionally, we introduce a Condition Prior Preservation Loss (CPPL) to address the issues of language and control drift caused by fine-tuning VLM and ControlNet on few-shot data. Our approach leverages large vision-language models (LVM) in combination with priors from specific input images to generate avatars that accurately reflect the input's appearance, while also allowing for editing via text prompts.

Extensive experimental results, e.g. Figure 4, on the HaveFun, AvatarBooth, and our own datasets demonstrate that PFAvatar surpasses state-of-the-art methods for avatar reconstruction and editing using multiple in-the-wild images. Additionally, As shown by Figure 5, we have also shown a strong generalization ability to the anime characters. We believe our endeavors would enhance the practical significance of this research area, paving a new way for human avatar reconstruction and real-world applications.

## 2 RELATED WORK

### 2.1 TEXT AND IMAGE-GUIDED 3D AVATAR GENERATION

Numerous studies have investigated methods for reconstructing clothed humans from visual inputs, such as multi-view images (Lin et al., 2024; Saito et al., 2019; Peng et al., 2021) or full-body monocular video (Weng et al., 2022; Li et al., 2020). Recently, a growing body of work has focused on generating human avatars guided by language descriptions. Early research in this area employed CLIP embeddings (Hong et al., 2022b) to shape rough body outlines. More recent approaches (Wang et al., 2023a; Liao et al., 2023; Kolotouros et al., 2023; Huang et al., 2023c; Cao et al., 2023a;

Hong et al., 2022b) have achieved finer geometric detail and texture for clothed individuals, or even multiple subjects, by leveraging large-scale text-to-image models and Score Distillation Sampling (SDS) (Wang et al., 2022; Poole et al., 2022).When subject images are available, they are utilized alongside text to fine-tune pretrained models (Ruiz et al., 2023) and improve accuracy through the use of re-projection losses (Yang et al., 2024; Huang et al., 2023b;a; Gao et al., 2023). While standard SDS frameworks typically require several thousand iterations, recent approaches (Chen et al., 2024) have accelerated the process through one-step generation based on image inputs. However, all image-based methods rely on precise human pose estimation (Pavlakos et al., 2019)to establish correspondences between the input image and the generated 3D avatar. Therefore, these approaches require images with clean backgrounds, standard body poses, and uncropped full-body views. PFAvatar overcomes the limitations of traditional methods. This makes it ideal for handling unconstrained, everyday photos from personal photo. By avoiding the need for precise pose estimation and geometric regularizers, ControlAvatar offers greater flexibility and can process a wide variety of real-world images without the strict constraints seen in other models.

## 2.2 FINETUNING OF DIFFUSION MODELS

In recent years, with the increasing interest in the text-to-image domain, pioneering researchers have begun exploring methods for personalizing text-to-image models using photos of specific subjects. Work on model customization introduces new concepts through fine-tuning (either partial or whole) of pre-trained networks (Avrahami et al., 2023; Jain et al., 2022; Kumari et al., 2023; Liu et al., 2024; Ruiz et al., 2023). Other research re-purposes diffusion models for new tasks (Fu et al., 2024; Ke et al., 2024; Kocsis et al., 2024). One representative work is DreamBooth (Ruiz et al., 2023), which uses a rare token to represent a specific subject or style, while preventing overfitting through a prior preservation loss. Another approach, textual inversion (Gal et al., 2022), generates a new embedding for the input concept and optimizes this embedding vector with a few photos to enable subject-driven image generation. LoRA (Hu et al., 2021)introduces a method for fine-tuning large language models by freezing the pre-trained model weights and injecting learnable rank decomposition matrices into the layers of the Transformer network (Vaswani et al., 2023). Despite these methods achieving laudable results with common objects, the abundance of prior information inherent in the human body poses challenges. This hinders the incorporation of such prior control when fine-tuning on human images. Consequently, consistency may diminish when integrating with controllers like ControlNet (Zhang et al., 2023b).

## 2.3 POSE-FREE RECONSTRUCTION IN THE WILD

In our study, "pose" encompasses both camera positioning and body articulation. The camera pose is vital for accurate 3D reconstruction because it aligns 3D geometry with 2D imagery (Mildenhall et al., 2021). However, determining camera pose from in-the-wild images is particularly difficult due to the uncontrolled nature of these environments. To address errors in camera estimation, some approaches incorporate joint optimization of both the object and camera parameters (Xia et al., 2022; Wang et al., 2021; Lin et al., 2021). Other methods rely on precomputed geometric cues (Meuleman et al., 2023; Fu et al., 2023; Bian et al., 2023) or use learning-based techniques for camera estimation (Zhang et al., 2024a; Wang et al., 2023c;b). Estimating body pose from in-the-wild images is particularly difficult due to its much higher dimensionality compared to camera pose. While some approaches can reconstruct static scenes from such images, even under challenging lighting and background conditions (Sun et al., 2022; Martin-Brualla et al., 2021), these methods are not suitable for handling articulated objects like human bodies. Based on our understanding, the work most pertinent to addressing our issue involves PuzzleAvatar (Xiu et al., 2024),Avatar-Booth (Zeng et al., 2023) and SIFU (Zhang et al., 2024c). PuzzAvatar and Avatarbooth create animatable 3D avatars from text descriptions and can also produce customized avatars using just a few phone photos or character designs generated by diffusion models.Unlike AvatarBooth, we do not use two diffusion models(Dual Model Fine-tuning), nor do we fine-tune with the original DreamBooth (Ruiz et al., 2023). In the reconstruction stage, they use the standard Score Distillation Sampling (Poole et al., 2022). The key difference with PuzzleAvatar is that their PuzzleBooth method is based on "BreakA-Scene" (Avrahami et al., 2023), which shows that jointly learning multiple concepts significantly boosts performance, possibly because this facilitates global reasoning when multiple regions are simultaneously generated. And in the reconstruction stage, they employ Noise-Free Distillation Sampling (NFDS) Katzir et al. (2023), an improved version of Score Distil-

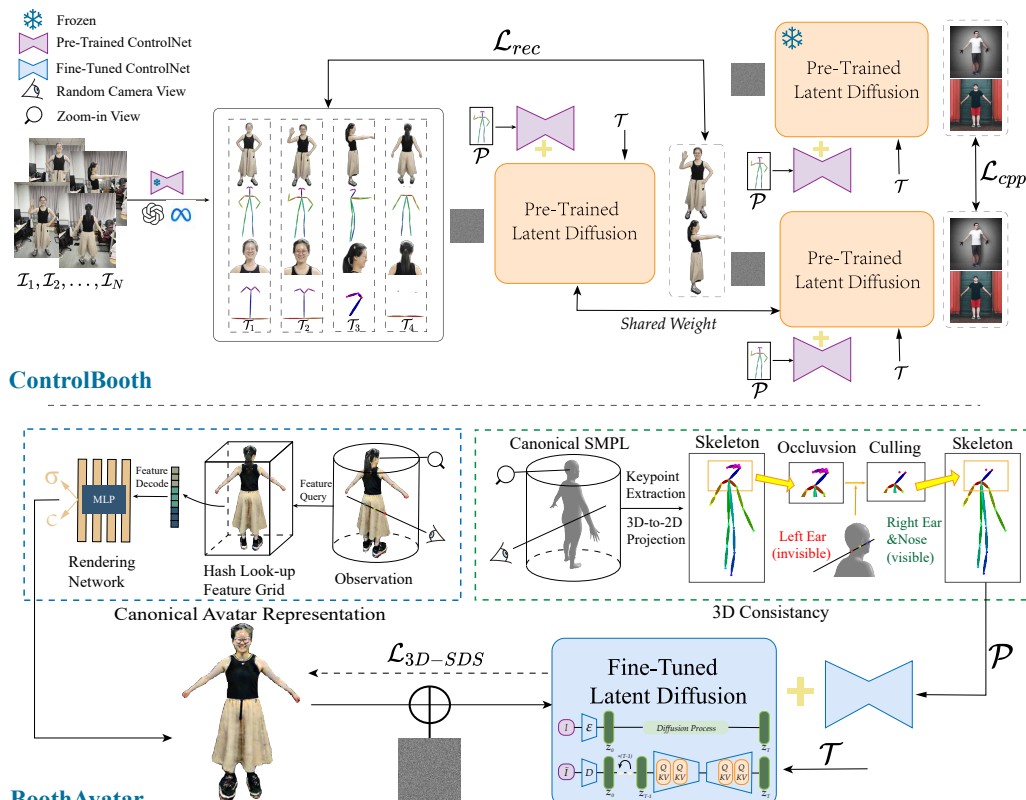

Figure 2: **Overview of our PFAvatar pipeline.** Our pipeline is primarily divided into two stages: **ControlBooth** and **BoothAvatar**. In the ControlBooth stage, we focus on fine-tuning text-to-image diffusion models and ControlNet for subject-driven generation, based on the collected images. During the BoothAvatar stage, we utilize the Fine-Tuned model obtained from the previous phase, employing multi-view 3D-consistent score distillation sampling to create a 3D avatar. For the network architectures of Latent Diffusion and ControlNet, please refer to Section A.1.

lation Sampling, though this sampling method does not take human-related prior information into account. While our approach shares similarities with PuzzleAvatar in decomposing the subject in the image to extract key information, we go further by incorporating human pose information during the decomposition process. Furthermore, as single-image pose-free reconstruction is a special case of multi image pose-free reconstruction, we selected the state-of-the-art work, SIFU (Zhang et al., 2024c), for single-image human reconstruction for comparison. By integrating ControlNet (Zhang et al., 2023b) to incorporate pose priors, enhances its ability for personalized generation. In the reconstruction stage, we utilize 3D-consistent Score Distillation Sampling(SDS) (Huang et al., 2023c) based on the Fine-Tuned Latent Diffusion to guide the sampling process, further improving the reconstruction performance.

## 3 METHOD

This section introduces the PFAvatar framework, which processes a set of in-the-wild images $\{\mathcal{I}_1, \mathcal{I}_2, \ldots, \mathcal{I}_N\}$ of a real person or anime character, and reconstructs a 3D avatar that faithfully captures both geometry and appearance. As shown in Fig. 2, the framework is divided into two primary stages. In the first stage, namely **ControlBooth** (Sec. 3.1), we fine-tune a Text-to-Image (T2I) (Rombach et al., 2021) and ControlNet (Zhang et al., 2023b) model to extract avatar features (pose, appearance) from the input images. In the second stage, namely **BoothAvatar** (Sec. 3.2), the fine-tuned T2I model is used as guidance to optimize the 3D avatar in the form of Neural Radiance Fields (NeRF) (Mildenhall et al., 2021) via 3D-consistent Score Distillation Sampling (SDS) (Huang et al., 2023c).

### 3.1 CONTROLBOOTH: INJECTING AVATAR PRIOR TO T2I AND CONTROLNET MODEL

#### 3.1.1 DREAMBOOTH FINETUNING ON CONTROLNET

Our first task is to finetune a personalized T2I and ControlNet model that can generate images of this avatar with varied poses. A T2I diffusion model (Saharia et al., 2022; Rombach et al., 2022; Ramesh et al., 2022) $\mathcal{D}_\theta(\epsilon, \mathbf{c})$ takes as input an initial noise $\epsilon \sim \mathcal{N}(0, 1)$ and a text embedding $\mathbf{c}_t = \Theta(\mathcal{T})$ for a given prompt $\mathcal{T}$ with a text encoder $\Theta$ and generates an image that follows the description of the prompt. However, it is difficult to exert fine-grained control in the generated images. Dream-Booth (Ruiz et al., 2023) proposes a simple yet effective approach to personalize a T2I diffusion model by fine-tuning the network on a small set of in-the-wild captures $\{\mathcal{I}_1, \mathcal{I}_2, \ldots, \mathcal{I}_N\}$. Briefly, DreamBooth uses the following diffusion loss function $\mathcal{L}$ (Eq. 1) to fine-tune the T2I model:

$$\mathcal{L} = \mathbb{E}_{\epsilon, t, \mathbf{c}_t} \left[ w_t \left\| \mathcal{D}_\theta(\alpha_t I_i + \sigma_t \epsilon, \mathbf{c}_t) - I_i \right\|_2^2 \right],\tag{1}$$

where $t \sim U[0, 1]$ denotes the time step in the diffusion process and $w_t, \alpha_t$, and $\sigma_t$ are the corresponding scheduling parameters.

**Applying DreamBooth on ControlNet**. While this method (Ruiz et al., 2023; Avrahami et al., 2023) of fine-tuning models has been employed in AvatarBooth (Zeng et al., 2023) and PuzzleA-vatar (Xiu et al., 2024) for 3D avatar generation, we find that their effectiveness is often compromised due to the significant pose variations of avatars. We aim to harness these diverse priors of avatar poses $\{\mathcal{P}_1, \mathcal{P}_2, \ldots, \mathcal{P}_N\}$ to enhance personalization capabilities. ControlNet (Zhang et al., 2023b) suppose $\mathcal{F}(\cdot; \Theta)$ is such a trained neural block, with parameters $\Theta$, that transforms an input feature map $x$ into another feature map $y$ as $y = \mathcal{F}(x; \Theta)$, where $x$ and $y$ are usually 2D feature maps, i.e., $x \in \mathbb{R}^{h \times w \times c}$ with $\{h, w, c\}$ as the height, width, and number of channels in the map, respectively. Thus, we input the pose $\mathcal{P}_i$ as a feature map $x$ into ControlNet, while other data is fed into T2I diffusion model for simultaneous fine-tuning to enhance personalized control capabilities. After the meticulous setup described above, we can proceed with training using the loss function $\mathcal{L}_{\text{rec}}$ from Eq. 2 as follows:

$$\mathcal{L}_{rec} = \mathbb{E}_{\epsilon, t, \mathbf{c}_{t_i}, \mathbf{c}_{p_i}} \left[ w_t \left\| \mathcal{D}_\theta(\alpha_t I_i + \sigma_t \epsilon, \mathbf{c}_{t_i}, \mathbf{c}_{p_i}) - I_i \right\|_2^2 \right],\tag{2}$$

where the $i$-th image-space condition $\mathcal{P}_i$ encodes a feature space conditioning vector $\mathbf{c}_{p_i}$, and $\mathbf{c}_i$ represents its corresponding text conditioning vector. Numerous existing works (Chen et al., 2023; Huang et al., 2023b; Liao et al., 2023) indicate that the view prompt aids in reconstruction. Therefore, we set a corresponding $\mathbf{c}_{t_i}$ for each image to enhance the model's performance. We provide a detailed description of the model architecture and implementation of the ControlBooth in A.1.

#### 3.1.2 CONDITION PRIOR PRESERVATION LOSS (CPPL)

**Motivation**. When fine-tuning with a small set of images, there is a risk of reducing the variability in the output poses and views of the avatar (e.g., snapping to the few-shot views). In addition, as shown in Figure 3 row 2, we also observed language drifting and reduced output diversity when combining ControlNet with DreamBooth fine-tuning.

**CPPL**. To address these issues, we propose a condition-based prior preservation loss, which promotes diversity, counters language drift, and helps maintain control capabilities. Specifically, we generate data $\mathcal{I}_{pr} = \mathcal{D}_\theta(\mathbf{z}_{t_1}, \mathbf{c}_{prt}, \mathbf{c}_{prp})$ using the ancestral sampler on the frozen pre-trained T2I diffusion model with random initial noise $\epsilon \sim \mathcal{N}(0, 1)$ and conditioning vectors $\mathbf{c}_{pr} := \Gamma(f_t(\mathcal{T}_{pr}))$ and $\mathbf{c}_{prp} := \mathcal{F}(\mathcal{P}_{pr})$. Here, $f_t$ is used to convert the prompt $\mathcal{T}_{pr}$ into the corresponding text embedding, while $\Gamma$ represents the text encoder that transforms it into the corresponding text conditioning vectors. Additionally, $\mathcal{F}$ is a neural block that converts the output 2D map $\mathcal{P}_{pr}$ into $\mathbf{c}_{prp}$. The form of $\mathcal{L}_{\text{cppl}}$ is given by Equation 3:

$$\mathcal{L}_{cppl} = \mathbb{E}_{\epsilon, t, \mathbf{c}_{prt_i}, \mathbf{c}_{prp_i}} \left[ \lambda w_t' \left\| \mathcal{D}_\theta(\alpha_t \mathcal{I}_{pr_i} + \sigma_t \epsilon, \mathbf{c}_{prt_i}, \mathbf{c}_{prp_i}) - \mathcal{I}_{pr_i} \right\|_2^2 \right],\tag{3}$$

where $\mathcal{L}_{\text{cppl}}$ is the condition prior-preservation term that supervises the model with its own generated images, and $\lambda$ controls the relative weight of this term. Figure 2 illustrates the model fine-tuning with the class-generated samples and the condition prior-preservation loss. Ultimately, our overall computational loss is shown in Equation 4:

$$\mathcal{L}_{\text{total}} = \mathcal{L}_{\text{rec}} + \mathcal{L}_{\text{cppl}}.\tag{4}$$

As shown in Figure 3 row 3, our introduction of $\mathcal{L}_{\text{cppl}}$ can effectively overcome the degradation of control capabilities. Typically, we set $\lambda$ to 1 during training.For details on data generation related to CPPL, please refer to Section A.1.

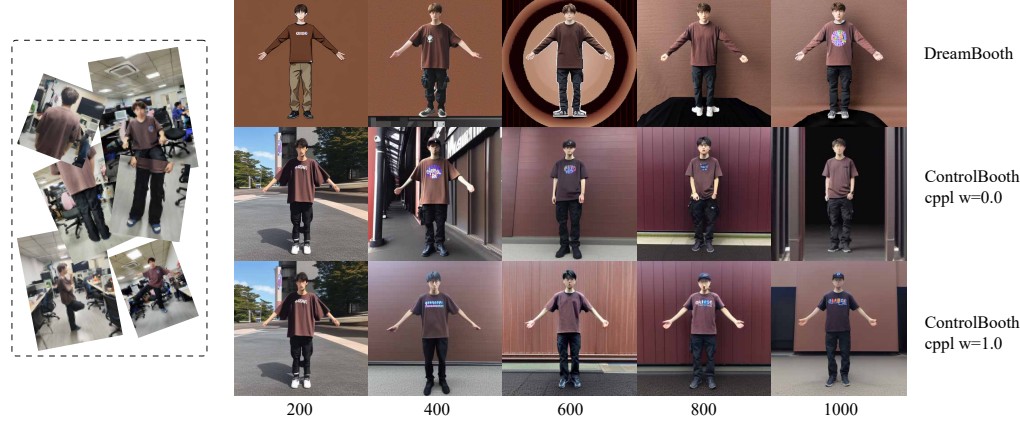

Figure 3: **Encouraging diversity while maintaining control through condition prior-preservation loss(CPPL).** Utilizing the fine-tuning strategy of Naive DreamBooth (Row 1) to generate images with new poses may introduce color discrepancies, significantly reducing consistency. Simply fine-tuning T2I(Row 2) and ControlNet may lead to overfitting on the context of the input image and the subject's appearance (e.g., pose). CPPL (Row 3) serves as a regularizer, mitigating overfitting while promoting diversity and maintaining control.

## 3.2 BOOTHAVATAR: RECONSTRUCT AVATAR VIA FINE-TUNED MODEL

**3D Representation.** We chose NeRF as our 3D representation. Neural Radiance Fields (NeRF) (Barron et al., 2021; Müller et al., 2022; Mildenhall et al., 2021) are widely used as 3D representations for text-to-3D generation (Guo et al., 2023; Lin et al., 2023b), and are parameterized by a trainable multilayer perceptron (MLP). To render an image, rays $\mathbf{r}(k) = \mathbf{o} + k\mathbf{d}$ are sampled, where $\mathbf{o}$ represents the camera position and $\mathbf{d}$ is the direction, both done on a per-pixel basis. The MLP takes these ray samples as input and predicts the density $\tau$ and color $\mathbf{c}$. The final pixel color is computed by approximating the volume rendering integral using numerical quadrature as follows:

$$\hat{C}_c(\mathbf{r}) = \sum_{i=1}^{N_c} \Omega_i \cdot (1 - \exp(-\tau_i \delta_i)) \, \mathbf{c}_i, \tag{5}$$

where $N_c$ refers to the number of sampled points along each ray, and $\Omega_i = \exp\left(-\sum_{j=1}^{i-1} \tau_j \delta_j\right)$ is the accumulated transmittance, with $\delta_i$ being the distance between consecutive sample points.

**SMPL-guided Initialization.** To accelerate NeRF optimization and provide a robust initial input for extracting insightful guidance from the diffusion model, we pre-train NeRF using an SMPL mesh. The SMPL model can be set in the canonical pose, as utilized in our approach to prevent self-occlusion, or in any preferred pose for creating posed avatars (Cao et al., 2024). Specifically, we render the image $\mathcal{I}_{\text{s}}$ of the SMPL model from a randomly sampled viewpoint and minimize the mean squared error (MSE) loss between the NeRF-rendered image $\mathcal{I}_{\text{r}}$ and the image $\mathcal{I}_{\text{s}}$

$$\mathcal{L}_{\text{MSE}} = \frac{1}{N} \sum_{i=1}^{N} \left(\mathcal{I}_{\text{r}}(i) - \mathcal{I}_{\text{s}}(i)\right)^2. \tag{6}$$

Empirical evidence reveals that SMPL-guided NeRF initialization significantly enhances both geometry and convergence speed during avatar generation.

**3D-consistent Score Distillation Sampling.** To fully leverage our pose fusion capabilities, we incorporate additional 3D-aware conditioning images to refine SDS (Huang et al., 2023c) for achieving 3D-consistent NeRF optimization. Specifically, an additional conditioning image $c$ is integrated into

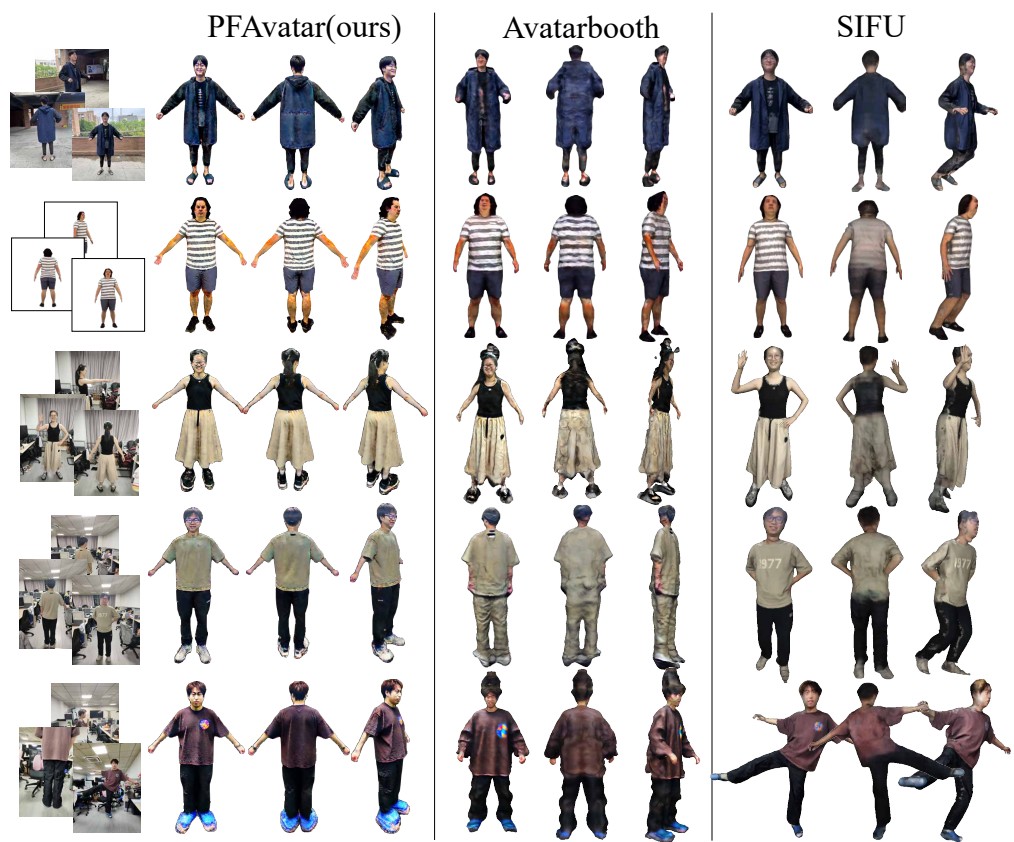

Figure 4: **Qualitative Comparison I: Real Person Dataset.** Visual results on three distinct subjects, employing two baseline techniques, AvatarBooth and SIFU alongside our method (PFAvatar), clearly showcase superior 3D consistency and subject fidelity in our approach compared to either baseline technique.

Equation 7 for the computation of the SDS gradient:

$$\nabla_{\boldsymbol{\theta}} \mathcal{L}_{\text{SDS}}(\phi, \mathbf{x}) = \mathbb{E}_{t,\boldsymbol{\epsilon}} \left[ w(t) \left( \boldsymbol{\epsilon}_{\phi} \left( \mathbf{x}_t; y, t, c \right) - \boldsymbol{\epsilon} \right) \frac{\partial \mathbf{z}_t}{\partial \mathbf{x}} \frac{\partial \mathbf{x}}{\partial \boldsymbol{\theta}} \right], \qquad (7)$$

where conditioning image $c$ can consist of one or a combination of skeletons, depth maps, etc. $w(t)$ is a weighting function dependent on the timestep $t$, and $y$ represents the associated text prompt. In practice, we choose skeletons as the type of conditioning image due to their provision of minimal image structure priors, which facilitate complex avatar generation. To ensure 3D-consistent guidance, the viewpoint of the conditioning image must align with that of NeRF's rendering. For avatar generation, we employ human SMPL models to generate these conditioning images.

**Zoom-in View for Head-Part.** To improve the quality of the avatar's facial structure, we implement a zoom-in view for the head. Specifically, we perform additional view sampling of the avatar's head with a probability that enhances facial clarity. By adopting this importance-based strategy, we can accelerate our training speed while simultaneously improving the quality of reconstruction.

## 4 EXPERIMENT

In this section, we perform comprehensive experiments to evaluate our method. We compare the performance of our method with state-of-the-art related methods and conduct ablation studies to validate the effectiveness of our designs. For more details on the experimental setup, please refer to Section A.1.

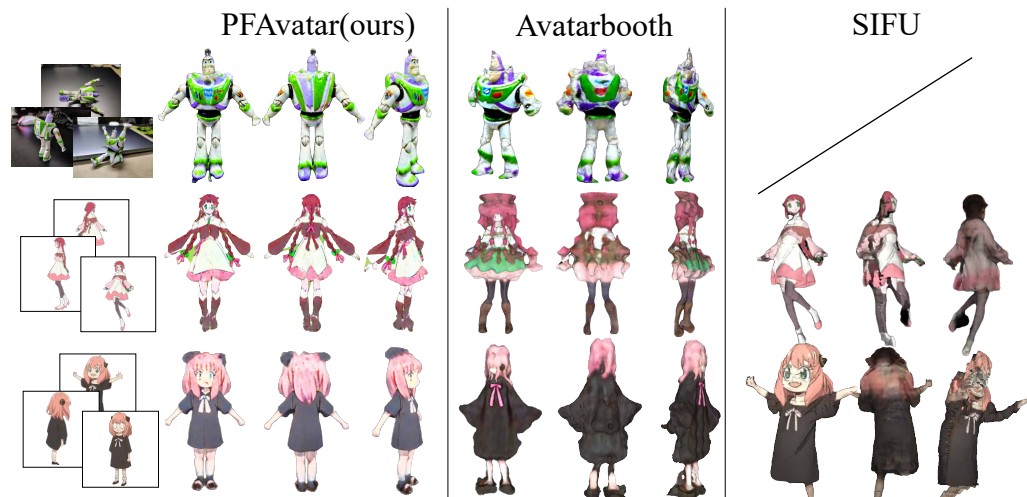

Figure 5: **Qualitative Comparison-II: Anime Character Dataset.** We compare our method with AvatarBooth and SIFU for appearance-customized reconstruction. Our method consistently achieves superior reconstruction quality and more faithful subject fidelity compared to all other approaches.

### 4.1 OVERVIEW

**Dataset Generation.** We perform experiments using 3 benchmark datasets: the real person dataset from Have-Fun (Yang et al., 2024) and the AvatarBooth dataset (Zeng et al., 2023) as well as our custom-built dataset containing both real people and anime characters. The Have-Fun dataset includes a variety of scenes with diverse character poses, though these scenarios are generated in a laboratory setting. The AvatarBooth dataset with varying poses, backgrounds, and camera angles presents significant reconstruction challenges. To demonstrate the efficacy of our approach, we developed our own dataset comprising real-person and character data. The former evaluates performance on real-person data, while the latter assesses the effects on character data. For more experimental results, please refer to Section A.2.

**Metrics.** We perform a quantitative evaluation on the dataset discussed earlier, focusing on an essential criterion: subject fidelity, which refers to how well the subject details are preserved in the generated images. Our evaluation leverages the DINO (Caron et al., 2021), CLIP-I, and CLIP-T metrics. CLIP-I measures the average cosine similarity between the CLIP (Radford et al., 2021) embeddings of generated and real images. The DINO metric computes the average cosine similarity between the ViT-S/16 DINO embeddings of the generated and real images. Meanwhile, CLIP-T assesses prompt fidelity by calculating the average cosine similarity between the text prompt and the corresponding image CLIP embeddings. As these CLIP metrics can only approximately gauge the quality and subject fidelity of the generated 3D assets. Specifically, the models generated by our method and previous works are first rendered into 1000 images from 25 different viewpoints. Subsequently, we compute the average metric. To ensure a more equitable evaluation, we additionally conduct user studies to compare various outcomes.

**Baselines.** PuzzleAvatar (Xiu et al., 2024) is the most similar concurrent work. Besides PuzzleAvatar, the work most relevant to ours is AvatarBooth (Avrahami et al., 2023), which we have chosen as one of our baselines. Since single image-to-3D is our special case, we have also selected SIFU (Zhang et al., 2024c) as an alternative baseline.

### 4.2 QUALITATIVE EVALUATIONS

Figure 4 and 5 show sample results of our approach in comparison to those of AvatarBooth and SIFU baselines. We demonstrate that our method surpasses all these works due to our designs. For further experimental comparisons, please refer to Section A.2.

**Comparison on real images.** We conducted a detailed qualitative comparison of each type of method using our chosen three categories of datasets of real persons. As depicted in Fig 4, Our

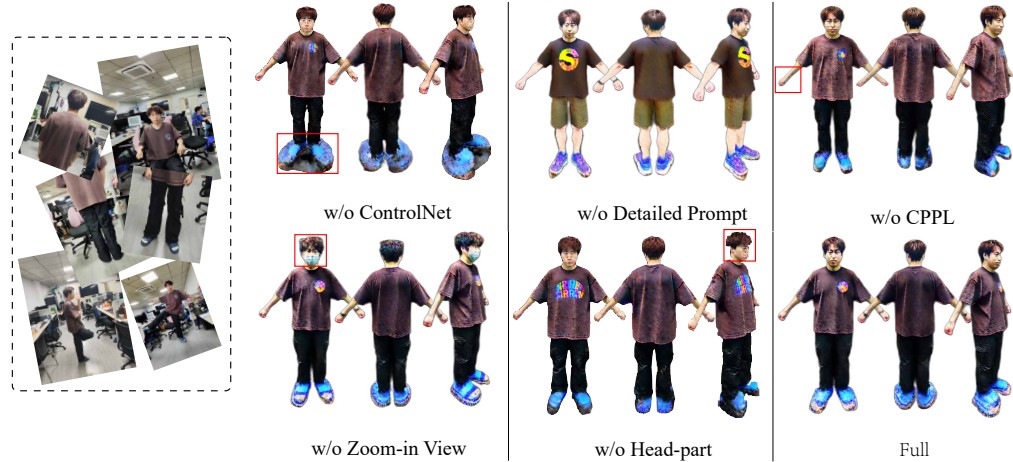

Figure 6: **Qualitative results of the ablation study.**

Table 1: **Quantitative comparisons** using DINO, CLIP-I, and CLIP-T on AvatarBooth, SIFU, and ControlAvatar generations demonstrate that renderings from our 3D model outputs more precisely capture the text prompts and image subjects.

| Method | CLIP-I↑ | | DINO↑ | | CLIP-T↑ | |
|---|---|---|---|---|---|---|
| | body | head | body | head | body | head |
| PFAvatar(Our) | **0.8922** | **0.9152** | **0.7772** | **0.8517** | **0.3136** | **0.2823** |
| Avatarbooth | 0.8533 | 0.8837 | 0.6778 | 0.7869 | 0.2907 | 0.2588 |
| SIFU | 0.8404 | 0.8970 | 0.7174 | 0.8371 | 0.2879 | 0.2748 |

method has various advantages over SIFU and AvatarBooth. SIFU works reasonably well in the front view (such as the person in the first row). It frequently introduces inconsistencies between the reconstructed front view and the hallucinated back view. In contrast, by handling all views with identity consistency, we enhance the coherence across different perspectives. Although AvatarBooth and our approach both utilize similar 3D representations, AvatarBooth employs two separate diffusion models to control the face and body. However, it relies on a single prompt for injection across all views and uses only vanilla SDS for guidance during reconstruction, resulting in lower-quality 3D avatars. In contrast, we individually process each view, injecting features independently into each image. By incorporating ControlNet to leverage avatar pose priors, we enhance identity consistency. During the reconstruction phase, our 3D-SDS further exploits pose priors to achieve superior results.

**Comparison on anime characters.** In this experiment, we qualitatively validate our ability to generate characters from anime character styles of data, conducting specific tests on anime-style datasets. As depicted in Figure 5, ControlAvatar has various advantages over SIFU and AvatarBooth. We find that even for the challenging anime character dataset—featuring complex clothing, unusual poses, and incomplete bodies—our method consistently achieved superior reconstruction quality and more faithful subject fidelity compared to all other approaches.

### 4.3 QUANTITATIVE EVALUATION

**Metric Evaluation.** Table 1 shows DINO, CLIP-I, and CLIP-T metrics for SIFU, AvatarBooth, and our PFAvatar Reconstruction. Results clearly demonstrate significantly higher scores for the PFAvatar results, indicating better 3D consistency, image subject fidelity, and text prompt alignment.

**User Study.** We carry out user studies to compare with the aforementioned state-of-the-art methods. Twenty-five volunteers are presented with 20 examples to assess these methods across four dimensions: (1) 3D Consistency, (2) Subject Fidelity, (3) Prompt Fidelity, and (4) Face Fidelity. They are asked to select the option that performs best among the given methods. The final ratings in 2 clearly demonstrate that PFAvatar is significantly favored over the baselines regarding 3D consistency, subject fidelity, face fidelity, and prompt fidelity.

Table 2: **User Study.** Users display a marked preference for our PFAvatar over AvatarBooth and SIFU in terms of 3D consistency, subject fidelity, face fidelity, and prompt fidelity.

| Method | 3D Consistency | Subject Fidelity | Prompt Fidelity | Head Fidelity |
|---|---|---|---|---|
| PFAvatar (ours) | **93.1%** | **88.3%** | **90.5%** | **92.6%** |
| Avatarbooth | 1.4% | 2.2% | 1.1% | 1.2% |
| SIFU | 7.5% | 10.5% | 9.4% | 7.2% |

Table 3: **Quantitative comparisons of ablation study.** Subject fidelity (DINO, CLIP-I) and prompt fidelity (CLIP-T, CLIP-T-L) ablation comparison.

| Method | CLIP-I ↑ | | DINO ↑ | | CLIP-T ↑ | |
|---|---|---|---|---|---|---|
| | full-body | head | full-body | head | full-body | head |
| Full | **0.8922** | **0.9152** | **0.7772** | **0.8517** | **0.3136** | **0.2823** |
| w/o ControlNet | 0.8741 | 0.8703 | 0.7259 | 0.8022 | 0.2807 | 0.2529 |
| w/o Detailed Prompt | 0.8457 | 0.9054 | 0.7391 | 0.8081 | 0.2414 | 0.2613 |
| w/o CPPL | 0.8820 | 0.8909 | 0.7593 | 0.8532 | 0.2680 | 0.2632 |
| w/o Zoom-in View | 0.8812 | 0.8531 | 0.7359 | 0.7620 | 0.2654 | 0.2484 |
| w/o head-part | 0.8631 | 0.8612 | 0.7486 | 0.8574 | 0.2792 | 0.2434 |

**Ablation Study.** In this section, we further conduct a set of experiments to evaluate the effectiveness of our designs. The comparison of metrics among these methods is illustrated in Table 3, while their qualitative comparison is shown in Figure 6. By introducing ControlNet, we have mitigated color bias. Compared to the coarse prompt description strategy of DreamBooth, providing detailed descriptions (Section A.1) for avatars has improved subject fidelity. Without CPPL, the character's skeleton can easily lose control, resulting in poorer generation quality, particularly in areas like the arms. By incorporating a zoom-in view for the head int BoothAvatar stage and introducing head-part data in the ControlBooth stage, we have enhanced the quality of the head region.

## 5 CONCULUSION

**Limitations**.
PFAvatar still has many limitations. As it builds on ControlBooth and SDS without incorporating reprojection terms, certain hallucinations are inevitable as shown in Figure 7, where the poses are too challenging for the avatar reconstruction. This could be improved by using improved pre-trained models and the adoption of image-based reprojection techniques.

**Conclusion**. In this paper, we introduce PFAvatar, a novel approach for reconstructing and editing avatars from multiple in-the-wild images. First, we extract avatar features such as pose and appearance using a Vision-Language Model (VLM) and ControlNet. Then, to capitalize on the pose-fusion prior, we utilize pose-conditioned 3D-Consistent Score Distillation Sampling (3D-SDS) to reconstruct a high-quality 3D avatar. To address the issues of language and control drift that may arise from fine-tuning VLM and ControlNet with few-shot data, we propose the Condition Prior Preservation Loss (CPPL). Experiments demonstrate the effectiveness of our method.

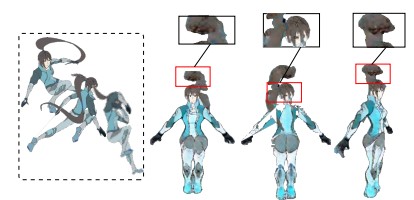

Figure 7: **Faliure Case.** For avatars with complex clothing and poses, relying solely on the SDS method may lead to the generation of hallucinations.

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

## A APPENDIX

### A.1 DEATILS OF THE PFAVATAR

**Model architecture of the ControlBooth.** This section presents the model architecture and implementation of the pre-trained model used during the ControlBooth phase. The structure of the network we train is derived from ControlNet, which includes the Latent Diffusion Model and ControlNet. Unlike the training method employed in ControlNet, we perform fine-tuning on the text encoder, U-Net, and ControlNet simultaneously. For the Latent Diffusion Model, we choose the Stable Diffusion Model V1.5 (Rombach et al., 2021). For ControlNet (Zhang et al., 2023b), we selected control_v11p_sd15_openpose as our pre-trained model.

**Dataset collection**. All the collected images undergo the following preprocessing step. The $\mathcal{I}_{head}$ images are provided by the user or extracted from the $\mathcal{I}_{body}$ images. To obtain a detailed description of each image, we will employ GPT-4V to analyze each one and extract the various features of the human body, such as upper clothing, lower clothing, etc, as well as the general direction the person is facing at that moment. Specifically, we will ask GPT-4V (GPT, 2023) to determine whether the person in the image is real, whether they are a known figure, the person's gender, accessories, hair (length, color, and other attributes), upper clothing (length, color, and other attributes), and lower clothing (length, color, and other attributes).

**Designing Detailed Prompts for Few-Shot Personalization.** During the training process, we discovered that detailed prompts for few-shot personalization significantly enhance the quality of human few-shot personalization. Specifically, rather than labeling all input images of the subject as "a [identifier] [class noun]," where [identifier] is a unique identifier linked to the subject and [class noun] is a coarse class descriptor of the subject (e.g., person, anime character, etc.), we instead use GPT-4V in advance to obtain detailed descriptions for each image. Ours queried Prompt as fellows:

*Analyze the provided all images, analyze the character's posture and facial expression at this time. If the person's demeanor cannot be analyzed at this time, only the facial expression at that time is given. Describe the gender of the character, and if it is a famous anime character or person, give the name of the anime character or person at the same time. The features that need to be identified are facial features (if there are facial ornaments, such as glasses, etc. on the face, corresponding descriptions need to be given), hairstyle, shoes, and clothing. For these features, you need to describe their specific length, color, style, etc. At the same time, the character's orientation at this time is given, which can be one of the following four situations: side view, front view, overhead view, or back view. Note that because you need to specify a character, you must add [identifier] to indicate a specific character. If there is only the head of the character in the picture at this time, please mark "Head," and please give some corresponding descriptions of the characteristics of the head.*

Extensive experimental evaluation, detailed text descriptions lead to higher quality during the training process.

**Data Augmentation for CPPL.** We perform random sampling of an SMPL human avatar to obtain the corresponding 2D openpose map $\mathcal{P}_{pr}$. In our experiments, we randomly sampled 250 different poses. Based on the azimuth angle at each pose, we also derived the direction $d$ for the prompt $\mathcal{T}_{pr}$. Using the pre-trained LDM and ControlNet models, we then generated 1,000 images for data augmentation.

**Implementation Details.** We implemented PFAvatar using the PyTorch framework with Diffusers (von Platen et al., 2022) and Threestudio (Guo et al., 2023). In the ControlBooth stage, we developed the algorithm for a fine-tuned diffusion model using the Diffusers library. In the BoothAvatar stage, we utilized Threestudio to create 3D avatars. Our training and inference were conducted on a single NVIDIA RTX 4090 GPU for all our results. The entire training process for both stages takes approximately 1 hour, with the ControlBooth stage taking 10 minutes and the BoothAvatar stage 50 minutes.

## A.2 MORE RESULT

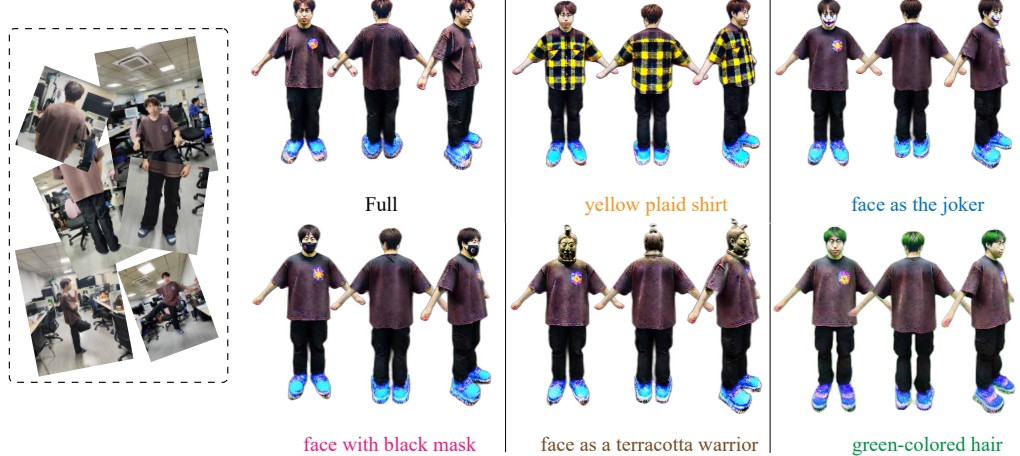

Figure 8: **More results on text-guided editing.**

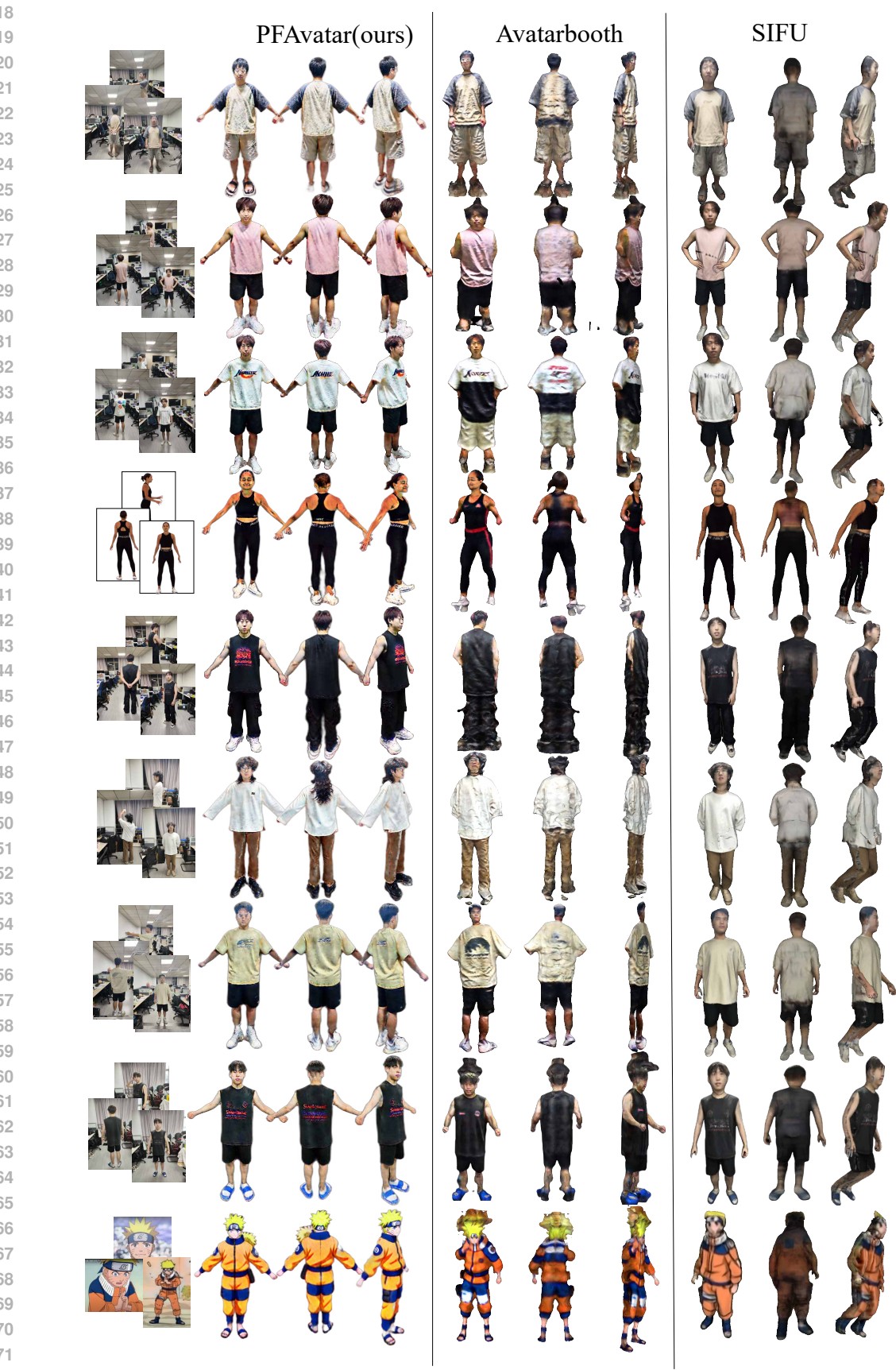

Figure 9: **More qualitative results of our self-collected datasets with both real human and anime characters.**