# OpenReview forum: "PFAvatar: Avatar Reconstruction from Multiple In-the-wild Images"
_ICLR.cc/2025/Conference — ICLR 2025 Conference Withdrawn Submission_

### Official Review · Reviewer_CXgz · 2024-11-01

**Soundness:** 2
**Presentation:** 1
**Contribution:** 2
**Rating:** 3
**Confidence:** 4

**Summary:**

- The paper investigates the reconstruction of human avatars from multiple in-the-wild images.
- The proposed method takes multiple casually captured images as input and outputs a 3D avatar.
- The approach is divided into two stages, named ControlBooth and BoothAvatar.
- In the first stage, the authors propose using ControlNet with pose as a condition, adopting a Dreambooth-style approach to finetune the diffusion model to generate subject-specific human avatar images.
- In the second stage, the authors use NeRF as a 3D representation, aiming to distill avatar information through SDS. SMPL is used for NeRF initialization and to obtain conditional skeletons for controllable avatar generation.
- The authors propose several improvements in conditioning losses.
- The paper demonstrates both qualitative and quantitative results across multiple datasets.

**Strengths:**

- The paper addresses a relatively interesting problem and attempts to combine more conditions based on Dreambooth to achieve a finetuning diffusion model method with fewer pose constraints.
- The extensive experiments reflect the authors' significant effort.

**Weaknesses:**

- The writing quality of the paper needs improvement.
  - The use of quotation marks should be reviewed, such as around "pose" on line 142.
  - The names "ControlAvatar" and "PFAvatar" should be consistently used throughout the paper.
- The methodology description is unclear.
  - The explanation in lines 223-224 lacks logical coherence and does not introduce the loss designed in Dreambooth to prevent performance degradation. Equation 1 seems more like a general T2I diffusion loss.
  - Section 3.1.1's explanation of combining Dreambooth and ControlNet does not clearly differentiate it from AvatarBooth.
    - Could it be summarized that using RGB directly for Dreambooth finetuning might be affected by pose, so 2D pose-guided ControlNet is added to perform finetuning on pretrained stable diffusion in a Dreambooth manner?
    - The difference between CPPL and Dreambooth's PPL lies in whether the sample generation involves ControlNet.
  - The introduction of NeRF in section 3.2 could be improved, as such a "general" 3D representation does not need repeated mention, especially since Equation 5's symbols are not specifically used later. Such content might be better suited for appendices or supplementary materials.
  - It remains unclear how SDS is integrated with NeRF during computation, and the text does not clearly explain this, nor does it clarify if Figure 2's BoothAvatar uses the previously trained ControlBooth.
- The experimental results are not very convincing.
  - The reproduced baselines and the newly proposed method, although not flattering, can be described as equally suboptimal.
  - The qualitative results of the proposed method show noticeable artifacts, such as in the feet (shoes) of the human figures, and there is a lack of explanation and analysis even if these artifacts are tolerable.

**Questions:**

- Does the BoothAvatar stage utilize the finetuned diffusion model from the first stage (ControlBooth) for SDS?
- How is background inconsistency handled during SDS? Is there foreground-background segmentation, or is there 3D-based filtering using NeRF to get the human subject?
- Can the generated result be something other than A-pose? Is it related to using A-pose SMPL during pre-training of nerf? If so, what is the significance of using non-A-pose during the first stage of data collection and learning DreamBooth?

---

### Official Review · Reviewer_4FM3 · 2024-11-02

**Soundness:** 2
**Presentation:** 1
**Contribution:** 1
**Rating:** 3
**Confidence:** 5

**Summary:**

The work PFAvatar presents a method for reconstructing 3D avatars in a canonical pose from multiple images. However, the method is not technically solid and does not produce good avatar reconstruction results. Additionally, the experiments conducted in the paper have several issues, which will be elaborated in the weaknesses and questions section. Furthermore, the manuscript contains several grammatical errors, indicating it is not yet ready for submission. For these reasons, I recommend rejecting the submission and encourage the authors to improve the quality of their work and resubmit.

**Strengths:**

The authors present several reconstruction results, including real-world humans and anime characters. Although the quality of these reconstructions could be improved, the results demonstrate that the proposed method has the potential to generalize across different domains, such as anime characters.

**Weaknesses:**

The weaknesses can be summarized into three main aspects:

Presentation: There are several severe and obvious presentation issues. While I will omit the grammatical problems, I will highlight the major presentation concerns:
- Line 119: "ControlAvatar" is used without definition. I assume this was the previous name of PFAvatar. Such an oversight is significant and suggests a lack of attention to detail by the authors.
- Line 160: The phrase begins with ". And in ...," which is another presentation issue that would not be present if the manuscript had been thoroughly proofread.
- Line 971: The figure title overlaps with the page number, indicating formatting issues.

Contribution: The paper's contributions are unclear and have limited novelty.
- limited task novelty: The task of monocular full-body image-based reconstruction is well-established. The method presented uses more input images and requires extensive time for ControlNet fine-tuning and SDS optimization to achieve a 3D representation, which seems inefficient.
- limited method novelty: Line 088 claims that 3D-SDS and CPPL are technical contributions, yet these appear more like incremental tweaks rather than substantive novelties.
- unsatisfied reconstruction results: he reconstruction results (e.g., the last row of Fig. 4) show unrealistic large shadows, and a strange black contour is visible on nearly all rendered PFAvatar reconstructions. In the supplementary video (11s - 13s), there is evidence of an empty volume in the reconstructed NeRF, further highlighting quality issues.

Experiments: The experimental setup and choice of baselines are confusing.
- Full-body images input: All presented reconstructions are based on full-body input images, which complicates the problem setting. Typically, personal album photos are casual and often contain only partial body shots (as in the PuzzleAvatar setup by Xiu et al.). State-of-the-art methods already handle 3D avatar reconstruction well from full-body images with efficient, feed-forward inference and lower GPU requirements. Therefore, proposing a task of reconstructing 3D avatars from multiple full-body images does not represent a novel contribution.
- Lack of comparison with PuzzleAvatar: The evaluation does not include results from casual, part-body photos, making it difficult to assess the effectiveness of the proposed method. Additionally, there is no comparison with PuzzleAvatar (Xiu et al.), which follows a similar SDS-based 3D representation optimization pipeline. Without this comparison, it is unclear if the proposed method is state-of-the-art in this context.

**Questions:**

The paper lacks a detailed breakdown of the inference running costs, which should include:

- Running time: Specifically, the time required for SDS-based optimization. This is important to understand the efficiency of the proposed method.
- GPU VRAM usage: The amount of GPU memory consumed during inference should be specified, as high VRAM requirements can be a limiting factor for practical application.
- Cost for GPT-4V-generated text descriptions: If GPT-4V is used to generate text descriptions, the associated costs should be included to provide a complete picture of the overall inference cost.

---

### Official Review · Reviewer_93rn · 2024-11-04

**Soundness:** 2
**Presentation:** 2
**Contribution:** 2
**Rating:** 5
**Confidence:** 4

**Summary:**

The paper introduces a novel framework called PFAvatar for reconstructing and editing 3D avatars from multiple in-the-wild images. This paper leverages VLM and T2I models as personalized priors to avoid the need for explicit per-pixel correspondences and camera pose estimation. Specifically, this paper presents a two-stage framework for avatar reconstruction. Firstly, ControlBooth finetunes a T2I and ControlNet model on in-the-wild to extract avatar features such as pose and appearance. Secondly, BoothAvatar uses the fine-tuned T2I model to optimize the 3D avatar with 3D-consistent SDS loss. Further, the paper introduces the CPPL module to address the drifting problem caused by finetuning.

**Strengths:**

1. This combination of VLM and ControlBooth is unique and tailored to the specific needs of avatar reconstruction from in-the-wild images, which is a significant departure from traditional methods that require precise pose estimation and geometric regularizers.

2. The CPPL module to mitigate issues of language and control drift during fine-tuning is also an interesting and effective contribution.

3. The paper is well-structured and clearly presents the methodology, experiments, and results.

**Weaknesses:**

1. Insufficient comparison. This paper only shows the comparison with Avatarbooth and SIFU. More single-image based avatar generation and 3D generation methods need to be included for comparison, such as PSHuman[1] and InstantMesh[2].

2. How does the method get the poses for the finetuning of the ControlNet? Is the method robust enough with inaccurate poses?

[1]Li, Peng, et al. "PSHuman: Photorealistic Single-view Human Reconstruction using Cross-Scale Diffusion." arXiv preprint arXiv:2409.10141 (2024).\
[2]Xu, Jiale, et al. "Instantmesh: Efficient 3d mesh generation from a single image with sparse-view large reconstruction models." arXiv preprint arXiv:2404.07191 (2024).

**Questions:**

Please see the weakness.

---

### Note · Authors · 2024-11-14

I have read and agree with the venue's withdrawal policy on behalf of myself and my co-authors.